# Prostate Cancer: A Journey Through Its History and Recent Developments

**DOI:** 10.3390/cancers17020194

**Published:** 2025-01-09

**Authors:** Hamza Mallah, Zania Diabasana, Sina Soultani, Ysia Idoux-Gillet, Thierry Massfelder

**Affiliations:** Regenerative NanoMedicine, Centre de Recherche en Biomédecine de Strasbourg, Fédération de Médecine Translationnelle de Strasbourg (FMTS), UMR_S U1260 INSERM and University of Strasbourg, 67085 Strasbourg, France; hamza.mallah@etu.unistra.fr (H.M.); z.diabasana@unistra.fr (Z.D.); yidouxgillet@unistra.fr (Y.I.-G.)

**Keywords:** prostate cancer, surgery, radiation therapy, hormone therapy, immunotherapy, resistance, personalized medicine

## Abstract

Along with lung, breast, and colon cancers, prostate cancer is among the four cancers with the highest global incidence with 1.5 million new cases and 400,000 deaths per year in 2022. PC is thus the world’s second most frequent cancer and the fifth leading cause of cancer-related death among men in 2022. Despite significant advances in the management of advanced PC during the last few decades, and since androgen was found to play a crucial role in PC progression in the mid-twentieth century, metastatic PC is currently considered incurable. Relevant preclinical tools that are representative of cancer heterogeneity are absolutely necessary to help design new therapeutic approaches. Even if progress is being made in this area, several questions still remain unresolved. Here, we summarized the history, the evolution, and the new developments in the management of PC. We also provide a comprehensive understanding of future directions of PC research, considering the molecular characteristics of the patients in a personalized medicine perspective.

## 1. Introduction

Prostate cancer (PC) ranks sixth among men’s cancer-related deaths worldwide and is the most frequently diagnosed malignancy in men [1]. In 2022, there were 1,466,680 newly diagnosed cases of this illness worldwide, resulting in 396,792 annual deaths [2]. PC is the most often diagnosed cancer worldwide, accounting for over 50% of all cancer diagnoses (112 of 185) [3]. The adult human prostate is structurally divided into central, transition, and peripheral zones [4]. The majority of prostate tumors start in the outermost peripheral zone [5]. Basal or luminal prostate epithelial cells are potential initiators of PC; with genetic manipulation, they can potentially produce high-grade lesions akin to adenocarcinomas [6]. The TMPRSS2 gene has a role in luminal differentiation. The most frequent chromosomal abnormality found in PC is a gene fusion between the oncogenic transcription factor ERG and TMPRSS2. This gene fusion drives carcinogenesis in over 50% of patients with PC [5,7].

The majority of PCs are low-grade, low-risk, and rarely aggressive. They also tend to grow slowly. Most of the time, there are no early or beginning symptoms, but late symptoms can include bone pain, exhaustion from anemia, paralysis from spinal metastases, and renal failure from bilateral ureteral obstruction. PSA testing and transrectal ultrasound (TRUS)-guided prostate tissue biopsies are the main methods used for diagnosis, while PSA testing for screening is still debatable because the PSA test is useful for finding little cancers. Furthermore, a lot of cancers discovered by PSA testing grow so slowly that they probably pose no threat to life [8,9,10].

Age, related health issues, tumor histology, and malignant extent all have an impact on tumor formation [11]. Five percent of men with distant metastases (often in multiple sites) are diagnosed with the disease, and fifteen percent of men with PC are diagnosed with locoregional metastases. Men with late-stage PC (distant metastases) have a dismal five-year overall survival rate of only 30% [12,13]. Localized PC just affects the prostate organ and is possibly treatable [14].

PC metastases are mainly linked to hematogenous dissemination to the stroma of the bone marrow and/or to the spread to locoregional lymph nodes. In bone tissue, metastatic lesions are seen in about 80% of cases. The majority of individuals with metastatic PC eventually develop castration-resistant PC (CRPC), a cancer that is resistant to androgen deprivation therapy (ADT). These characteristics are the main contributors to PC mortality and morbidity [15]. Eventually, therapy- and castration-resistant PC, which has no more effective treatment options and is regarded as an end-stage disease, develops from metastatic CRPC [16].

Furthermore, PC exhibits remarkable heterogeneity and can be further classified into multiple intermediate clinical states, each of which may benefit from a distinct therapeutic approach [17]. For instance, patients with indolent or low-risk tumors typically follow active surveillance regimens; those with localized disease typically undergo radiotherapy and radical prostatectomy surgery; and those with aggressive or metastatic cancer typically receive a combination of multiple targeted therapies, including hormonal therapy, radiotherapy, chemotherapy, and immunotherapy [18].

The immunotherapeutic medication sipuleucel-T, the alpha-emitter bone-seeking radioisotope radium-233, the two androgen signaling inhibitors abiraterone and enzalutamide, and the chemotherapy medication cabazitaxel are among the current treatment choices [19].

In general, PSA screening refers to a structured program or policy where men, typically aged 50 to 75, are invited to participate in a PSA test and possibly a digital rectal examination as part of a surveillance approach to identify prostate cancer [20]. A digital rectal exam is a standard examination that men often have to screen for rectal or PC. In fact, in the absence of a PSA rise, a digital rectal anomaly is suspected in around 5% of cases of PC that have been detected. But normal digital rectal results do not eliminate PC [21]. Since most prostate tumors are found in the periphery, a digital rectal examination will reveal them if their volume is more than 0.2 milliliters [22].

The diagnosis and surveillance of PC can be greatly aided by MRI [23]. Results from randomized controlled trials indicate that compared to routine transrectal biopsies, MRI-directed biopsies detect around twice as many clinically relevant malignancies (grades 2–4) [24]. Also, PCA3, a noncoding messenger RNA (mRNA) specific to the prostate, has been discovered to be overexpressed in over 90% of all prostate cancers. Previous research has reported the utilization of PCA3 RNA quantification in post-DRE urine. The Progensa PCA3 assay (Progensa Test Kit, Hologic, Marlborough, MA, USA) is a diagnostic test that has been approved by the FDA and is recommended for use in males 50 years of age and older who have an increased serum PSA and previously negative results from a prostate biopsy [25].

Typically, a prostate biopsy is carried out if cancer is suspected. To ensure that every part of the prostate is sufficiently sampled, TRUS guidance is usually always used for this procedure. Taking two specimens from each of the three regions (base, mid-gland, and apex) on both sides is the most widely utilized pattern [26]. The goal is to more precisely pinpoint the tumor’s position and extent. By using a transperineal biopsy, the risk of infection is reduced from approximately 1% to nearly 0%. This form of prostatic biopsy is becoming more and more widespread, particularly in Europe where it is the recommended and preferred procedure [27]. For individuals with high-risk prostate cancer, PSMA PET-CT (positron emission tomography–computed tomography) provided a more accurate diagnosis than conventional imaging. Results from retrospective single-center studies that used histopathology as the standard of reference for pelvic lymph node staging prior to prostatectomy suggest that PSMA PET-CT may be more accurate than CT or MRI in this regard. PSMA PET-CT performed better than CT or MRI in a trial involving 130 patients with intermediate-to-high-risk PC, with a diagnosis accuracy of 0.83 (0.76–0.91) versus 0.69 (0.59–0.79) [28,29].

In this review, we summarize the types of PC, the diagnosis, and the different treatments, and we retrace the history of therapies (localized vs. metastatic) with developments to arrive at current therapies with the most effective and innovative of these therapies.

## 2. The Major Signaling Pathways in PC

Androgen stimulation is essential for the proliferation, survival, and early detection of PC in healthy prostate tissue [30]. In order to prevent prostate cancer from dying, the activated PI3K-Akt signal pathway is necessary [31]. Furthermore, a number of tumor-suppressive genes, including PTEN, are downregulated, absent, or altered in early PC [32]. PC is also promoted by the activation of the glucose transporter GLUT12, which is induced by correlation AR with CaMKK2 signaling [33]. Moreover, PC cell proliferation and invasion are facilitated by asparaginyl endopeptidase through the PI3K/AKT signaling pathway [34].

NF-κB is often activated in prostate tumor cells as a result of elevated receptor levels, such as TNF, which significantly increase IκB degradation. In androgen-independent prostate tumors, NF-κB expression is increased at the mRNA and protein levels due to increased expression of interleukin 6 (IL-6) [35,36]. Also, the JAK/STAT pathway is developed in PC and activates the STAT gene, which in turn activates the anti-apoptotic and angiogenesis genes, causing the growth of cancer [37].

Wnt signaling is dependent on the ability to stabilize the multifunctional protein β-catenin via the Wnt/β-catenin pathway, which results in asymmetric cell division, differentiation, migration, and epithelial–mesenchymal interactions in PC [38,39].

The hedgehog (Hh) signaling pathway also plays a crucial role in prostate cancer development and progression. Activation of the Hh pathway, often through loss of Su(Fu), overexpression of sonic hedgehog protein, or mutations in pathway components, contributes to increased cell proliferation, enhanced invasiveness, resistance to apoptosis, and promotion of epithelial–mesenchymal transition (EMT) [40,41,42].

Both autocrine and paracrine signaling mechanisms are involved, sometimes in combination.

High levels of Hh target genes, such as PTCH1 and HIP, are found in over 70% of tumors with high Gleason scores, linking this activation to increased prevalence, malignant progression, poor prognosis, and higher mortality [40,41].

Targeting the Hh pathway has shown promise in treatment, with inhibitors like cyclopamine and vismodegib demonstrating antitumor effects by suppressing invasiveness, inducing apoptosis, and inhibiting EMT in castration-resistant prostate cancer models [40,43].

Additionally, botanical compounds such as genistein, curcumin, EGCG, and resveratrol may offer safer alternatives [44].

Combining Hh inhibitors with therapies like radiation, chemotherapy, or other molecular agents could improve treatment outcomes and help overcome drug resistance [41].

The BAD has been demonstrated to be phosphorylated and inhibited by the cAMP/PKA pathway, which results in PC resistance to apoptosis [45]. Moreover, CREB, or cAMP response element binding protein, can stimulate HDAC2 and GRK3, which in turn promotes neuroendocrine differentiation and angiogenesis [46,47]. Also, PC differentiation has been shown by a correlation between the hedgehog pathway (HH) and the pluripotency-inducing transcription factor SOX2 [48].

The family of proteins known as angiomotin (AMOT) was first discovered to bind angiostatin and control endothelial cell migration and tube formation [49,50]. A study showed that inhibition of the Hippo pathway, which causes YAP to translocate to the nucleus and increases the production of the YAP target gene BMP4, is one of the mechanisms underlying AMOTp80-mediated proliferation, so BMP4 expression stimulates PC cell growth [51].

The activation of the cathepsin A (CTSA) gene by the p38 MAPK pathway plays a significant role in the progression of PC [52].

The activation of the protein kinase B (AKT) signaling pathway caused by the overexpression of CHIP in DU145 PC cells led to a rise in cyclin D1 protein levels and a decrease in p21 and p27 protein levels, thereby promoting cell proliferation [53]. Targeting the metabolism of cancer cells, NFATc1 can introduce certain oncogene molecules, such c-myc, that promote cell division and growth [54].

In advanced PC, the inhibitory effects of TGF-β on cell cycle arrest and cell proliferation become resistant, most likely because the functional distribution of action between Smad-independent and SMAD-dependent signaling is disrupted. As a result, TGF-β transforms from a tumor suppressor to a tumor promoter, accelerating the spread of PC to metastasis [55]. Long noncoding RNAs (lncRNAs) promote PC by regulating the let-7a/TGF-β1/ SMAD signaling pathway [56].

## 3. Clinical Evolution of PC

Prostate cancer (PC) manifests in various forms, each with distinct characteristics and treatment approaches. This section outlines the major types of PC, from benign prostatic hyperplasia to advanced metastatic forms.

Benign prostatic hyperplasia, which is the non-malignant enlargement or hyperplasia of the prostate tissue, is one of the most frequent causes of LUTS in older men [57].

Localized PC is the first stage of PC. When this occurs, the cancer is limited to the prostate and is treated with radiation or surgery. Hormone therapy may also be employed on occasion. A biochemical recurrence could occur as it progresses, increasing the PSA. It may also develop into CRPC that is not metastatic [58,59].

nmCRPC cancer grows after hormone treatment. It might not have migrated to other body parts, according to scans.

Metastatic PC is the result of advanced PC. It extends to other bodily parts in addition to the prostate. Hormone therapy is no longer effective in stopping the growth of mCRPC. Other organs and tissues can also have cancer [60].

Oligometastatic PC refers to the formation of limited sites of distant dissemination following initial RP or radiation; there is ongoing disagreement over the nature of the first metastasis (de novo vs. recurrent) and the type of prior systemic therapy (hormone-sensitive versus castration-resistant) [61,62].

Metastatic PC is an advanced form of PC that has begun to grow outside of the prostate and is resistant to initial therapies like hormone therapy and surgery; it is known as mCRPC and its precursor, mHSPC [63].

Charles Huggins and Clarence Hodges established the hormone sensitivity of PC metastases in 1941. Achieving castrated testosterone levels through surgical procedures like bilateral orchidectomies or medical castration using ADTs like GnRH analogues was the cornerstone of treatment for mHSPC. ADT’s effects on the hypothalamic–pituitary axis reduce the amount of testosterone produced by the testicles, which produce 90–95% of all androgens. The most commonly used tactic is ongoing GnRH agonist therapy, which inhibits the generation of luteinizing hormone and, consequently, the synthesis of androgens in the testicles.

mCRPC, which is characterized by disease progression despite ADT with castrate testosterone levels, frequently develops in mHSPC treated with ADT. This can manifest in a number of ways, such as a persistent increase in blood levels of the PSA, the advancement of pre-existing illness, or the emergence of new metastatic deposits [64].

Neuroendocrine prostate cancer (NEPC) is an aggressive and rare subtype of prostate cancer, often associated with poor prognosis and limited treatment options. It can arise de novo or evolve from adenocarcinoma, particularly as a resistance mechanism to androgen receptor (AR)-targeted therapies in castration-resistant prostate cancer (CRPC).

NEPC is characterized by the loss of AR dependence, with low or absent AR expression, making it unresponsive to AR-targeted therapies. Common genomic alterations include the loss of tumor suppressors such as TP53 and RB1, along with epigenetic dysregulation involving EZH2. This subtype is known for its aggressive phenotype, rapid progression, high metastatic potential, and resistance to conventional therapies.

First-line treatment for NEPC typically involves platinum-based chemotherapy (cisplatin or carboplatin with etoposide), although responses are often short-lived [65,66].

Second-line therapy options include docetaxel, amrubicin, irinotecan, or combinations of platinum-based drugs, but progression-free survival is generally less than six months [65]. For patients with homologous recombination repair (HRR) gene alterations, PARP inhibitors may offer some benefit. Immune checkpoint inhibitors and other molecularly targeted therapies are currently under investigation [65,67].

Emerging strategies for treating NEPC include engineered exosomes targeting NEPC-specific surface antigens like CEACAM5 to deliver drugs that inhibit EZH2 and AR. Biomarker-driven approaches using liquid biopsies and genomic profiling are also being explored for early detection and personalized treatments [67,68]. Combination therapies, such as combining EZH2 inhibitors with immune checkpoint inhibitors or PARP inhibitors, are being investigated to target multiple pathways simultaneously. Preclinical models, including patient-derived xenografts, are being used to better understand the molecular drivers of NEPC and identify new therapeutic targets [67,69,70].

## 4. Classification of PC

PC classification is crucial for determining appropriate treatment strategies and predicting outcomes. This section outlines the TNM staging system and clinical categorization of PC.

The terms Tumor (T), Node (N), and Metastasis (M) in the TNM staging method relate to the tumor’s location and size (T), its dissemination to lymph nodes (N), and the metastasis dissemination to other body areas (M) [71]. T1a to T2c represent prostate tumors in the organ without extra-prostatic extension; T3 tumors are cancers that have spread outside of the prostate and into seminal vesicles or periprostatic fat. A tumor is classified as cT4 if it invades nearby structures, such as the bladder, rectum, levator muscles, or pelvic sidewall [72,73].

The involvement of regional lymph nodes (N) is graded as follows: Stage NX: no regional lymph nodes evaluated; Stage N0: metastasis of regional lymph nodes; and Stage N1: metastasis of regional lymph node(s). The most advanced category is chosen for distant metastasis (M) staging, which includes the following: Stage M0 denotes the absence of distant metastases, Stage M1 denotes distant metastases, and Stage M1a denotes distant metastases other than regional lymph nodes. Stage M1c denotes other site(s) with or without bone involvement; Stage M1b denotes metastasis to bone(s) [74,75,76].

In clinical practice, PC is most conveniently categorized as follows [77,78]:
Localized PC: T1/2/ early T3, N0, M0;Oligometastatic PC: Established T3 or T4, N0/1, M0;Metastatic cancer: M1 disease.


## 5. Progression to mCRPC

Progression to metastatic castration-resistant prostate cancer (mCRPC) involves several complex mechanisms. Androgen receptor (AR) alterations play a major role, with AR amplification occurring in 30–80% of CRPC cell lines, leading to hypersensitivity to low androgen levels. AR mutations also enable activation by non-androgen molecules, and AR splice variants can form heterodimers with wild-type AR, promoting castration-resistant growth [79,80]. Persistent androgen signaling is maintained through intratumoral androgen synthesis and adrenal androgen sources, both of which remain unaffected by androgen deprivation therapy (ADT). Changes in coactivators and corepressors, such as upregulation of FKBP51 and steroid receptor coactivators (SRC-1, SRC-2, SRC-3), also contribute to mCRPC progression, with SRC-3 overexpression linked to poorly differentiated, advanced prostate cancer [79]. Aberrant pathway activation, including PI3K/AKT signaling due to PTEN loss and the intertwining of MAPK and PI3K signaling with AR signaling, further promotes tumor growth. Epithelial–mesenchymal transition (EMT) is another key event, marked by the upregulation of growth factors (TGF-β, IGF-1) and signaling pathways (MAPK, PI3K), as well as the loss of epithelial markers and the acquisition of mesenchymal markers.

Epigenetic modifications, such as 5hmC modification at binding sites of key oncogenes, and DNA repair deficiencies contribute to mCRPC progression and may affect treatment response [81,82,83]. Despite initial responses to ADT, progression to mCRPC is often inevitable, making it critical to understand these mechanisms in order to develop targeted therapies and optimize treatment strategies for improved outcomes in advanced prostate cancer [24].

## 6. Tumor Heterogeneity

Prostate cancer exhibits significant tumor heterogeneity at multiple levels, which poses challenges for diagnosis, treatment, and research. Tumors from different patients often exhibit unique sets of genetic alterations, contributing to interpatient heterogeneity in prostate cancer. The Cancer Genome Atlas Research Network identified seven molecular subtypes of primary prostate carcinomas, each characterized by specific genetic alterations. In addition to interpatient heterogeneity, prostate cancer also displays intertumoral heterogeneity. In multifocal disease, separate tumor foci within the same prostate can have completely different genotypes [84,85].

Studies have shown that these different tumor foci often arise as clonally distinct lesions, with no shared driver gene alterations, further highlighting the complexity and diversity of prostate cancer at the genetic level. Intratumoral heterogeneity is also common, with individual tumor foci exhibiting subclonal diversity, and about 60% of tumors showing multiple subclones, which is associated with a higher rate of recurrence. DNA ploidy heterogeneity is also observed, with some studies finding differences in DNA ploidy classifications within tissue blocks and needle biopsy specimens [85,86]. This tumor heterogeneity complicates accurate diagnosis and grading, as sampling from different areas may yield varying results. It also has prognostic implications, as more genetically heterogeneous tumors are associated with worse disease outcomes. Additionally, heterogeneity poses challenges for targeted therapies since different areas of the tumor may respond differently to treatment, and it can drive drug resistance by promoting tumor evolution, particularly in response to intense hormonal therapy [17,60,86].

To address these challenges, strategies being explored include multi-region sequencing to capture the full spectrum of genetic alterations, the development of novel technologies to better characterize heterogeneity, and considering heterogeneity in the design of clinical trials and treatment strategies. Understanding and addressing tumor heterogeneity is crucial for improving prostate cancer management [17,85].

## 7. History of PC Treatment

The treatment of PC has evolved significantly over the past century, from its first documented case to modern therapeutic approaches. This section chronicles the key milestones in PC treatment, highlighting major discoveries and advancements in surgical, hormonal, and chemotherapeutic interventions.

The first case of PC was reported by surgeon J. Adams of The London Hospital in 1853. Adams’s diagnosis came from a histological examination. Adams described this ailment as “a very rare disease” in his study [87]. It is incredible how, 150 years later, PC has become a significant health concern [87].

Huggins and Clarence Hodges discovered the use of castration or estrogen therapy as a treatment for PC between 1930 and 1940 [88,89,90]. Castration and estrogen therapy can lead to thromboembolic damage and cardiovascular disease, and they are insufficient treatments for advanced PC. Thus, in the 1960s and 1980s, novel strategies were developed involving hormone therapies (anti-androgens, LHRH agonists, LHRH antagonists) to either prevent the generation of adrenal androgen or limit the interaction of androgens with target tissue [91]. In late 1960, it was discovered that cyproterone acetate inhibited the androgen receptor’s ability to bind either testosterone or dihydrotestosterone (DHT). Additionally, it was a progesterone agonist that binds to pituitary progesterone receptors to decrease LH secretion [92]. In 1970, it was found that flutamide can counteract the negative effects of cyproterone acetate. The US Food and Drug Administration (FDA) authorized it in 1989 for the treatment of PC, making it the first non-steroidal anti-androgen to undergo clinical testing. Later research produced bicalutamide and nilutamide, two more pure non-steroidal anti-androgens [87].

When these new medications were being developed, it became evident that patients with advanced PC could not be cured by orchiectomy, LHRH agonists, or anti-androgens alone [93]. In 2000, it was shown that androgen receptor pathway inhibitors (ARPIs) improve survival in metastatic castration-resistant PC (mCRPC), nmCRPC, and mHSPC in comparison to normal therapy [94].

ARPIs include abiraterone acetate, enzalutamide, apalutamide, and darolutamide [95]. Enzalutamide, aplutamide, and darolutamide are used to suppress AR activity [96]. But abiraterone specifically and permanently inhibits CYP17 (17 alpha-hydroxylase/C17,20-lyase), an enzyme expressed in testicular, adrenal, and prostatic tumor tissues that is necessary for the synthesis of androgens, and prevents the production of androstenedione and dehydroepiandrosterone (DHEA), two precursors of testosterone [97]. The first two medications to be approved in 2013 by the FDA for mCRPC were abiraterone and enzalutamide [98]. The results of the ARASENS study from 2022 have shown that darolutamide is the first ARPI to show a survival benefit when used in combination with docetaxel in mHSPC [99].

Surgical procedures such as prostatectomy were first employed in 1904 to treat localized PC [100]. With the invention of radium sources at the start of the 20th century, radiation therapy for localized PC first became available [100,101].

Cytotoxic chemotherapy was used in a few limited trials in the 1950s and 1960s for hormone-refractory PC [102]. The program claimed minimal toxicity and subjective improvement in the first national randomized research comparing 5-fluorouracil versus cytoxane versus conventional therapy in 1975 [103]. Later studies showed that chemotherapy might be used in addition to other treatments to treat advanced PC [104].

Mitoxantrone, a novel chemotherapy drug, was first made available in 1996 to treat mCRPC. However, it has been demonstrated that mitoxantrone solely improves palliative endpoints, not OS [105].

In 2004, docetaxel was shown to enhance OS either alone or in combination with estramustine in the TAX-327 and SWOG 99-16 trials [106]. The E3805 trial demonstrated in 2015 that docetaxel was also effective in treating metastatic hormone-sensitive PC (mHSPC) [99,107]. The only other chemotherapy drug, cabazitaxel, a taxane that is structurally related to docetaxel, has been demonstrated to enhance OS in metastatic PC [108]. In the 2010 TROPIC study, it was demonstrated to increase survival in mCRPC when used as a second-line treatment following docetaxel failure in comparison to palliative mitoxantrone [99,109].

## 8. Current Treatment for PC

Prostate cancer treatment varies significantly depending on the stage, risk group, and other factors such as PSA levels, Gleason score, and the patient’s overall health.

Figure 1 summarizes the various treatment options based on the clinical stage of prostate cancer, including localized prostate cancer, advanced prostate cancer, hormone-sensitive metastatic prostate cancer, and hormone-resistant metastatic prostate cancer.

### 8.1. Management of Localized PC

There is no consensus regarding the optimum management of localized disease. Patients need to be made aware of the advantages and disadvantages of each option. Patients should be provided with the chance to consult with both a radiation oncologist and urologist due to the variety of treatment choices available and their associated adverse effects. Patients should be warned that sexual dysfunction, infertility, and bowel and bladder issues can result from PC treatment.

For men who are not suitable for, or reluctant to receive, treatment with the intention of curing, surveillance combined with delayed hormone therapy for symptomatic progression is an alternative. Active surveillance refers to a close monitoring approach that usually involves PSA, repeat biopsies, and MRI, with the goal of maintaining curative treatment for patients exhibiting signs of disease progression. There are insufficient data to compare various active surveillance techniques [110]. Radiotherapy, low-dose-rate brachytherapy, and radical prostatectomy are available as treatments. From October 1989 to February 1999, 695 men with localized PC were randomly randomized to either watchful waiting or radical prostatectomy as part of the Örebro trial, which also collected follow-up data until 2017. As of 31 December 2017, 292 of the 348 men in the watchful-waiting group and 261 of the 347 men in the radical-prostatectomy group had died. PC was responsible for 110 deaths in the watchful-waiting group and 71 in the radical-prostatectomy group, leading to a relative risk of 0.55. This results in an absolute risk difference of 11.7 [111]. The PIVOT trial, which involved 731 North American men from 1994 to 2002, indicated that there was no significant advantage to surgery for the 296 patients in the low-risk subgroup, where the 12-year lethality probability was less than 3%. In fact, the findings suggest that surveillance may be more beneficial than surgery, as reflected in both PC-specific and overall mortality rates. Additionally, the overall death rate of about 50% at ten years underscores the presence of significant comorbidities among the recruited patients [110].

ProtecT is a phase III clinical trial that is prospectively randomized and compares active therapy, such as RP or RT, with active monitoring, which involves repeat biopsies in patients who have a PSA rise of more than 50% from the baseline value. The ProtecT trial randomly assigned 1643 men with localized prostate cancer to three treatment arms: active monitoring, radical prostatectomy (RP), and radiotherapy (RT). The active monitoring approach in ProtecT differed from traditional active surveillance, as PSA levels were measured every three months in the first year and twice yearly thereafter, with a rise of ≥50% in PSA triggering further review and potential testing. After 15 years of follow-up, the study found similar prostate-cancer-specific mortality across all groups (3.1% in active monitoring, 2.2% in RP, and 2.9% in RT). However, disease progression differed: metastases occurred in 9.4% of the active monitoring group, compared to 4.7% in RP and 5.0% in RT, while local progression was seen in 25.9% of the active monitoring group, versus 10.5% in RP and 11.0% in RT. Notably, 24.4% of patients in the active monitoring group remained alive without any prostate cancer treatment at the end of the follow-up period [112].

According to a clinical observation made by Huggins and Hodges in 1941, localized PCa growth may be managed by reducing the level of androgens by castration [113].

### 8.2. Oligometastatic PC

PC with three to five metastatic lesions is commonly known as the oligometastatic condition, which can be further categorized based on the time interval between the emergence of the metastases. With more sensitive molecular imaging approaches, metastasis-directed therapy for oligometastatic disease has gained significant interest and helped to reclassify nonmetastatic disease as metastatic disease [114,115].

Metastasis-directed therapy may increase survival, decrease the rate of biochemical recurrence, or lengthen the time to following therapy, according to three randomized phase 2 trials. Between 10 February 2012 and 30 August 2016, 99 patients were randomized, with 33 (33%) assigned to the control group and 66 (67%) to the SABR group (stereotactic ablative radiotherapy). In this study, two patients (3%) in the SABR group did not receive the assigned treatment and withdrew from the trial, as did two patients (6%) in the control group. The median follow-up duration was 25 months for the control group and 26 months for the SABR group. Median OS was 28 months for the control group, compared to 41 months for the SABR group [116,117].

### 8.3. Metastatic Hormone-Sensitive PC (mHSPC)

Metastatic PC was classified as either castration-sensitive or -resistant, denoting a stage of the illness that can be castrated early on or a stage that can no longer be treated with testosterone suppression [118].

#### 8.3.1. ADT (Adjuvant Therapy)

For patients with mHSPC, ADT is the usual treatment. It may include bilateral orchiectomy, LHRH agonists, or LHRH antagonists. ADT was the only available modality in earlier decades, but it is no longer the standard of care (SOC). The average survival time for mHSPC patients receiving ADT alone is around 42 months [119]. The introduction of combination therapy in 2015, also known as “intensification beyond androgen deprivation therapy” caused a paradigm change. Studies evaluating the effectiveness of adding docetaxel or androgen biosynthesis inhibition with abiraterone acetate and prednisone together with blockage of the androgen receptor with enzalutamide or apalutamide showed improved OS [118].

#### 8.3.2. ADT + Docetaxel

In the TAX 327 trial, docetaxel was the first cytotoxic drug to show an advantage for survival for advanced PC [120] (Appendix A). In 2015, the US-led CHAARTED study administered ADT with or without six cycles of docetaxel to 790 patients with mHSPC. This trial was the first to demonstrate that adding docetaxel to ADT resulted in a significant increase in OS by 13.6 months [107].

When combining androgen deprivation therapy with an androgen pathway inhibitor or docetaxel, clinical factors like disease volume, comorbidities, drug accessibility, and patient preference are usually taken into consideration [24].

#### 8.3.3. ADT + Abiraterone

In two phase III trials (LATITUDE, STAMPEDE), the addition of abiraterone to ADT showed improved OS compared with ADT alone [110]. Until the disease progressed, individuals in both trials were randomly assigned to receive ADT alone or in combination with abiraterone 1000 mg + prednisone 5 mg daily. The LATITUDE study, which enrolled from 12 February 2013 to 11 December 2014, randomly allocated 1199 patients with high-risk metastatic PC. Among these patients, 602 were placed in the placebo group and 597 in the abiraterone group if they had GS ≥8 or three or more visceral or bone metastases. With the addition of abiraterone to ADT, OS considerably improved (HR 0.62; 95% CI 0.51–0.76) [121].

A total of 1917 willing patients across 111 U.K. and 5 Swiss locations participated in the STAMPEDE study, which ran from 15 November 2011 to 17 January 2014. Of these, 957 received ADT alone and 960 received combination therapy. Of these patients, 95% had recently been diagnosed with a disease; 52% of the patients had metastatic disease, 20% had node-positive or node-indeterminate nonmetastatic disease, and 28% had node-negative, nonmetastatic cancer (40 months was the median follow-up period). In the combined group, 184 people died, while 262 people died in the ADT-alone group [122].

#### 8.3.4. ADT + Enzalutamide

A phase III trial conducted from March 2014 to March 2017 at 83 sites, managed by the NHMRC Clinical Trials Centre, enrolled 1125 men who were randomly assigned to receive either standard care (androgen deprivation therapy (ADT) along with a conventional non-steroidal anti-androgen like flutamide, benalutamide, or nilutamide) or enzalutamide (160 mg daily). This included 562 patients in the standard care group and 563 in the enzalutamide group.

The findings revealed that enzalutamide significantly enhanced both clinical PFS and PSA progression-free survival compared to OS. Specifically, the enzalutamide group recorded 174 PSA progression-free survival events, while the standard care group had 333 events. After three years, the event-free survival rates were 67% for those on enzalutamide and 37% for standard care. Regarding OS, the rates were 72% for the ADT-alone group and 80% for the enzalutamide group, indicating a survival benefit with enzalutamide, though the most notable improvement was observed in PSA progression-free survival [123].

#### 8.3.5. ADT + Apalutamide

The phase 3 TITAN study was designed by the sponsor, Janssen Research and Development. It was conducted at 260 sites across 23 countries between 15 December 2015 and 25 July 2017. A total of 527 patients received a placebo plus ADT, while 525 patients received apalutamide plus ADT. The patients were 68 years old on average. The apalutamide group had a 33% lower risk of death compared to the placebo group, with an overall survival rate of 82.4% at 24 months, versus 73.5% in the placebo group. Additionally, undetectable PSA levels were observed in 28.7% of patients receiving placebo, while this was seen in 68.4% of patients treated with apalutamide [124].

### 8.4. Nonmetastatic Castration-Resistant PC

Recently recognized as a disease state, high-risk nCRPC satisfies the following criteria: (i) PC verified by histopathology or cytology, (ii) rise in PSA in spite of castrated testosterone levels, (iii) PSA doubling time of 10 months or less, and (iv) no signs of metastatic illness, either past or present, as determined by whole-body radionuclide bone scan, computed tomography, or magnetic resonance imaging of the chest, abdomen, and pelvis. Patients with high-risk non-medical CRPC have shown improvements in PFS and OS when second-generation inhibitors like enzalutamide, apalutamide, or darolutamide are used to target the androgen receptor [118,125].

### 8.5. Metastatic Resistant PC

Most patients with metastatic PC will eventually develop worsening disease even after castration. Patients with metastatic castration-resistant disease who continue taking LHRH agonists while undergoing systemic therapy may benefit marginally from them in terms of survival [118].

#### 8.5.1. Chemotherapy

From October 1999 to January 2003, 674 eligible patients were enrolled in a SWOG 99–16 study. Among them, 338 were assigned to receive docetaxel and estramustine, while 336 were given mitoxantrone and prednisone. The findings indicated that patients treated with docetaxel and estramustine had a longer median overall survival compared to those receiving mitoxantrone and prednisone, with survival times of 17.5 months versus 15.6 months. In the docetaxel and estramustine group, the median time to progression was 6.3 months, compared to 3.2 months for the mitoxantrone and prednisone group. Moreover, 50% of patients treated with docetaxel and estramustine and 27% of those receiving mitoxantrone and prednisone showed a reduction in PSA levels of at least 50% (Appendix A) [106].

In a phase 3 trial (EFC6193; TROPIC) conducted across 146 centers in 26 countries between 2 January 2007 and 23 October 2008, a total of 755 patients were randomly assigned to treatment groups, with 378 receiving cabazitaxel and 377 receiving mitoxantrone. In addition to prednisone, patients received either mitoxantrone at 12 mg/m^2^ or cabazitaxel at 25 mg/m^2^ for a maximum of 10 cycles. As of 25 September 2009, there had been 279 deaths among patients treated with mitoxantrone and 234 among those treated with cabazitaxel. The median OS was significantly higher for the cabazitaxel group, at 15.1 months, compared to 12.7 months for the mitoxantrone group, reflecting a 30% reduction in the relative risk of death. Furthermore, the median PFS was 2.8 months for the cabazitaxel group, while it was 1.4 months for the mitoxantrone group. Cabazitaxel combined with prednisone demonstrates notable clinical antitumor activity, improving overall survival for patients with metastatic castration-resistant PC whose disease has progressed during or after docetaxel therapy (Appendix A) [109].

#### 8.5.2. AR-Targeted Therapy

Treatments for mCRPC with novel AR-targeted treatments are well established. In both the pre-docetaxel and post-docetaxel disease states, the COU-AA-301 phase III trial enrolled 1195 patients at 147 sites across 13 countries between 8 May 2008 and 28 July 2009. Eligible patients had metastatic castration-resistant PC that continued to progress despite treatment with docetaxel. In the study, 797 patients were randomly assigned to receive abiraterone acetate plus prednisone (abiraterone: 1000 mg once daily), while 398 patients received a placebo along with prednisone (placebo group: 5 mg twice daily). The median OS was significantly longer for the abiraterone group at 15.8 months compared to 11.2 months for the placebo group. The 95% median time to PSA progression was 8.5 months. The median PFS was 5.6 months for the abiraterone group, while the placebo group had a median of 3.6 months. Furthermore, the proportion of patients with a PSA response was significantly higher in the abiraterone group, with 235 (29.5%) of 797 patients responding compared to 22 (5.5%) of 398 patients in the placebo group (*p* < 0.0001). Abiraterone acetate significantly extends overall survival in patients with metastatic castration-resistant PC who have experienced disease progression following docetaxel treatment [126] (Appendix A). Additionally, abiraterone decreased skeletal-related events, enhanced pain management, and enhanced quality of life. The strong AR antagonist enzalutamide, which inhibits AR nuclear translocation and binding to androgen response elements on DNA, is licensed for use for similar indications as abiraterone [24].

#### 8.5.3. Radium-223

Radium-223, an alpha-emitting calcium mimetic, has a lower risk of hematologic problems because it binds to the microenvironment of sclerotic metastases with a far narrower range of irradiation than beta emitters. From June 2008 to February 2011, the phase III ALSYMPCA trial enrolled a total of 921 patients (614 in the radium-223 group and 307 in the placebo group) at 136 study centers in 19 countries, all included in the intention-to-treat analysis. Patients in the radium-223 group received a median of six injections, while those in the placebo group received five. The updated analysis showed a median overall survival of 14.9 months for the radium-223 group, compared to 11.3 months for the placebo group. This analysis confirmed a 30% reduction in the risk of death for those receiving radium-223 versus placebo. Furthermore, radium-223 significantly prolonged the time to the first symptomatic skeletal event, with a median of 15.6 months for the radium-223 group versus 9.8 months for the placebo group. Additionally, a greater proportion of patients in the radium-223 group had a response based on total alkaline phosphatase levels (≥30% reduction) compared to those in the placebo group (Appendix A) [127,128,129].

#### 8.5.4. Sipuleucel-T

An autologous cellular immunotherapy called sipuleucel-T is authorized for the treatment of mCRPC that is asymptomatic or very mildly symptomatic. It is made up of fusion protein PA2024, which is prostatic acid phosphatase connected to granulocyte macrophage colony-stimulating factor, in autologous antigen-presenting cells. In the phase III study APC8015, 127 patients were randomly assigned to receive three infusions every two weeks of either placebo (n = 45) or sipuleucel-T (n = 82). Patients in the placebo group were allowed to receive APC8015F, derived from frozen leukapheresis cells, as their disease progressed. At the time of the data analysis, 115 of the 127 patients had experienced disease progression, and all were monitored for 36 months to evaluate survival. The sipuleucel-T group had a median time to disease progression (TTP) of 11.7 weeks, compared to 10.0 weeks for the placebo group. The median survival for patients receiving sipuleucel-T was 25.9 months, while those on placebo had a median survival of 21.4 months. Furthermore, patients treated with sipuleucel-T exhibited an eight-fold increase in the median ratio of T-cell stimulation at eight weeks compared to their baseline levels (16.9 vs. 1.99). The treatment with sipuleucel-T was found to be well tolerated [130,131] (Appendix A).

#### 8.5.5. Poly (ADP-Ribose) Polymerase Inhibitors

Inhibiting poly (ADP-ribose) polymerase (PARP) has long been investigated as a potential treatment for ovarian and breast malignancies, particularly when BRCA1/2 or other germline DNA damage repair abnormalities are present. Recent large-scale multicenter studies showed that up to 11.8% of patients with advanced PC have germline abnormalities in DNA damage repair genes [132]. Roughly twenty-three percent of metastatic castration-resistant PCs have somatic or germline loss-of-function mutations in DDR genes, including ATM, CHEK2.34, BRCA2, and BRCA1 [133].

In the phase 3 PROfound trial from 2016 to 2020, the PARP inhibitor olaparib was studied in men with metastatic castration-resistant PC who had disease progression while receiving a new hormonal agent, such as enzalutamide or abiraterone. In this trial, 2792 patients were randomly assigned to receive olaparib or enzalutamide or abiraterone (control) at 206 sites and 20 countries. The median imaging-based PFS, as determined by independent review, was significantly longer for the olaparib group than for the control group, with medians of 5.8 months versus 3.5 months. Among patients who could be evaluated, the confirmed objective response rate was 22% (30 of 138) in the olaparib group and 4% (3 of 67) in the control group (odds ratio, 5.93; 95% CI, 2.01 to 25.40). After six months, 85% of patients in the olaparib group were free from pain progression, compared to 75% in the control group. In an interim analysis, with data maturity at 41%, the median OS was 17.5 months for the olaparib group and 14.3 months for the control group. Additionally, a PSA response was confirmed in 30% (73 of 243) of patients in the olaparib group, versus 10% (12 of 123) in the control group. Furthermore, 27% (41 of 153) of patients in the olaparib group and 10% (7 of 68) in the control group demonstrated conversion of circulating tumor cells. Olaparib was associated with longer progression-free survival and superior response measures, along with enhanced patient-reported outcomes, when compared to enzalutamide or abiraterone. This benefit was especially evident in patients who had disease progression while being treated with enzalutamide or abiraterone and who had genetic alterations affecting homologous recombination repair (Appendix A) [134].

#### 8.5.6. mTOR Inhibitors

In late-stage mCRPC, PTEN depletion is linked to an aggressive clinical course and is a target for pharmaceutical management. From 30 June 2017 to 17 January 2019, the phase 3 IPATential150 trial was conducted at 200 sites in 26 countries, screening 1611 patients for eligibility, of which 1101 (68%) were enrolled. Participants were randomly assigned to either the placebo–abiraterone group (554 patients, 50%) or the ipatasertib–abiraterone group (547 patients, 50%). PTEN loss, identified by immunohistochemistry, was found in 521 (47%) of the patients, and the baseline characteristics were generally balanced between the two treatment groups. In the placebo–abiraterone group, the median duration of placebo treatment was 14.0 months, while abiraterone treatment also lasted a median of 14.0 months. In the ipatasertib–abiraterone group, the median duration of ipatasertib treatment was 11.1 months, with the abiraterone treatment lasting a median of 14.2 months. Among those with PTEN loss, the median duration for the placebo group was 13.8 months (with abiraterone also at 13.8 months), while the median duration for the ipatasertib group was 11.3 months (with abiraterone at 14.3 months). A total of 143 patients (26%) in the placebo–abiraterone group and 126 patients (23%) in the ipatasertib–abiraterone group died, primarily due to disease progression. A study was conducted to evaluate the therapeutic efficacy of the AKT inhibitor ipatasertib. Individuals with mCRPC who were asymptomatic or just slightly symptomatic before treatment were eligible. When compared to abiraterone + placebo, the combination of ipatasertib and abiraterone improved PFS in patients with PTEN loss (18.5 months vs. 16.5 months). In men with PTEN-loss metastatic CRPC (mCRPC), a population characterized by a poor prognosis, the combination of AKT inhibition and androgen receptor signaling pathway blockade using ipatasertib and abiraterone offers a potential treatment strategy (Appendix A) [135].

#### 8.5.7. Immunotherapy

Immune checkpoint inhibitors have somehow revolutionized the therapy paradigm for many malignancies. Examples of these inhibitors include monoclonal antibodies that target CTLA4, the protein PD-1, or one of its ligands, PD-L1. It is still unclear, though, what role these treatments will play for unselected PC patients. Response rates of about 15% were observed in early-phase investigations with PD-1 and PD-L1 axis inhibition [136,137].

In the phase II KEYNOTE-199 study, 258 out of 394 patients screened were enrolled from 1 July 2016 to 31 March 2017. Patients with RECIST-measurable PD-L1-positive and PD-L1-negative diseases were assigned to cohorts 1 and 2, respectively, while those with bone-predominant disease were included in cohort 3. All participants received pembrolizumab at a dosage of 200 mg every three weeks for a maximum of 35 cycles. The median rPFS for cohorts 1, 2, and 3 was 2.1 months, 3.7 months, and 2.1 months, respectively. The median OS was 9.5 months for cohort 1, 7.9 months for cohort 2, and 14.1 months for cohort 3. The 12-month survival rates were reported as 41%, 35%, and 62% for cohorts 1, 2, and 3, respectively. When cohorts 1 and 2 were combined, the median rPFS and OS were both 2.1 months and 8.1 months, respectively. For all three cohorts together, the median rPFS was 2.1 months, and the median OS was 9.6 months. Pembrolizumab monotherapy shows antitumor activity with an acceptable safety profile in a subset of patients with RECIST-measurable and bone-predominant mCRPC previously treated with docetaxel and targeted endocrine therapy (Appendix A) [137,138].

#### 8.5.8. Prostate-Specific Membrane Antigen Theragnostic

PSMA is a desirable theragnostic target for paired imaging and treatment since it is expressed in the majority of prostate tumors and is upregulated in mCRPC. All tumor sites can have PSMA expression measured using PSMA PET-CT (with various radiolabels), and patients who meet the criteria can be provided PSMA-specific therapy [24].

Systemic PSMA-targeted radioligand treatment (RLT) was developed for mCRPC, introducing a new therapeutic approach. A tiny chemical called lutetium-177-PSMA-617 (LU-PSMA) attaches exclusively to PSMA, allowing β particle treatment to reach nearby tumor cells in CRPC. A positive diagnostic 68-Gallium PSMA PET scan is required in order to identify patients who will benefit from this molecular treatment.

Between June 2018 and mid-October 2019, a phase III study VISION enrolled 831 out of 1179 screened patients, who were randomly assigned to receive either 177Lu-PSMA-617 with standard care (the 177Lu-PSMA-617 group) or standard care alone (the control group) at 84 sites (52 in North America and 32 in Europe). The combination of 177Lu-PSMA-617 and standard care significantly increased both imaging-based PFS (median of 8.7 months compared to 3.4 months) and OS (median of 15.3 months versus 11.3 months). The addition of RLT with 177Lu-PSMA-617 improved both imaging-based PFS and OS for patients with advanced PSMA-positive metastatic castration-resistant PC when combined with standard care (Appendix A) [139].

#### 8.5.9. Combination Therapies

##### Enzalutamide with Talazoparib

For patients with untreated mCRPC, the ALAPRO-2 trial was a randomized, double-blind, phase 3 study conducted across 26 countries, comparing talazoparib plus enzalutamide to placebo plus enzalutamide as a first-line treatment for men from 7 January 2019 to 17 September 2020. A total of 805 participants were enrolled, with 402 assigned to the talazoparib group and 403 to the placebo group. Patients received either 0.5 mg of talazoparib or a placebo, alongside enzalutamide at 160 mg, taken orally once daily, after their tumor tissue was assessed for HRR gene alterations. The median follow-up for rPFS was 24.9 months in the talazoparib group and 24.6 months in the placebo group. In the primary analysis, the median rPFS for the talazoparib plus enzalutamide group was not reached (95% CI 27.5 months to not reached), while the median rPFS for the placebo plus enzalutamide group was 21.9 months. The combination of talazoparib and enzalutamide demonstrated a clinically meaningful and statistically significant improvement in rPFS compared to standard enzalutamide alone as a first-line treatment for patients with metastatic castration-resistant PC (mCRPC) (Appendix A) [140].

##### Abiraterone with Niraparib

From May 2019 to March 2022, the third phase of the MAGNITUDE study enrolled 423 patients. Among them, 212 were assigned to receive niraparib plus abiraterone, while 211 received a placebo plus abiraterone. In the BRCA1/2 subgroup, the rPFS showed a significant difference between the niraparib + abiraterone and placebo + abiraterone groups (16.6 months vs. 10.9 months). The combination of niraparib and abiraterone significantly enhanced rPFS in patients with HRR+ mCRPC compared to the standard treatment of abiraterone plus placebo. In the overall HRR+ cohort, the rPFS was significantly longer for the niraparib + abiraterone group than for the placebo + abiraterone group (16.5 months vs. 13.7 months). Both the US FDA and Health Canada have approved niraparib in combination with abiraterone, but with restrictions (Appendix A) [141].

##### Abiraterone with Olaparib

In the PROpel trial, 1103 patients were screened between 31 October 2018 and 11 March 2020. Out of these, 399 patients were randomly assigned to receive olaparib plus abiraterone, while 397 were assigned to placebo plus abiraterone. The trial included men with mCRPC who were at least 18 years old. For the patients whose data were censored, the median follow-up for OS was 36.6 months for the olaparib plus abiraterone group and 36.5 months for the placebo plus abiraterone group. The median OS for those receiving olaparib plus abiraterone was 42.1 months compared to 34.7 months for the placebo plus abiraterone group. In this final prespecified analysis, there was no significant difference in overall survival between the two treatment groups (Appendix A) [142].

## 9. Novel Agents and Clinical Trials for PC in 2023–2024

Recent advancements in prostate cancer treatment have led to the development of novel agents and ongoing clinical trials. This section highlights some of the promising approaches being investigated.

RDC medications (framework law on public health) modify particular targets and deliver cytotoxic or imaging chemicals to the target area of PC around normal tissue by using antibodies or radionuclide isotopes such as 177Lu, 111In, and 225Ac-J591 (a PSMA-targeted radionuclide therapy) [143].

Most individuals with nonmetastatic CRPC (M0CRPC) have demonstrated a significant decrease in prostate-specific antigen (PSA) levels when treated with ketoconazole or hydrocortisone in conjunction with radioactive 177Lu and 111In. In mCRPC, a novel triple therapy including 225Ac-J591 (a PSMA-targeted radionuclide therapy), pembrolizumab, and an androgen receptor pathway inhibitor (ARPI) showed a significant reduction in PSA response [143,144].

DART is used to attract and redirect immune effector cells in order to kill tumor cells or stop other signaling pathways by preventing the ligand or receptor from functioning. For example, Lorigerlimab, which maintains a maximum blockage of PD-1 while enhancing CTLA-4 blocking of dual expression, has been shown to promote antitumor effectiveness in mCRPC patients who are refractory to chemotherapy [145].

CAR-T cells bind to target proteins on tumor surfaces, which promotes T-cell expansion, releases cytokines, and releases anti-apoptotic proteins. BPX-601, a CAR-T example utilized in mCRPC, shows that only 14.3% (1/7) of patients showed disease progression, while 14.3% (1/7) of patients maintained stable disease (SD) for more than nine months [146,147].

AKTi, also known as ATP Competitive AKTi, is a pyrimidine and alkyl phospholipid molecule that binds to AKT and traps it to obstruct downstream signaling pathways. A phase III study has shown that administering capivasertib and docetaxel to patients with mCRPC increases overall survival [148,149].

The TRANSFORM trial is currently the largest PC screening trial in 20 years, comparing multiple screening options including fast MRI scans, genetic testing, and PSA blood testing [150]. The ENACT trial, now inactive, evaluated enzalutamide monotherapy and found a 46% reduced risk of prostate cancer progression compared to active surveillance [151]. Several ongoing phase 2 trials are investigating promising treatments, including Vudalimab (XmAb20717) for advanced gynecologic and genitourinary malignancies [152] and olaparib combined with an LHRH agonist as neoadjuvant therapy for high-risk localized prostate cancer [153]. Xaluritamig (AMG509), a STEAP1 × CD3 bispecific antibody, is being studied in a first-in-human trial for metastatic castration-resistant prostate cancer [154]. These trials represent ongoing efforts to improve prostate cancer screening, treatment, and management across various stages of the disease.

Appendix A provides an overview of both active and inactive clinical trials in PC, highlighting novel agents and approaches being investigated in 2023–2024.

## 10. siRNA-Based Therapy in PC

siRNA therapy has emerged as a promising approach in PC treatment due to its specificity in targeting and silencing cancer-associated genes.

Since its discovery, siRNA has shown significant promise in the treatment of many diseases, including cancer, because of its specificity in targeting and silencing genes associated with the etiology of cancer, such as those involved in immunosuppression, cell proliferation, and invasion [133]. Combining gene-specific siRNAs with other traditional treatments like radiation and chemotherapy has demonstrated beneficial results in the treatment of PC recently [134]. A number of investigations verified that siRNA targeting of particular genes might significantly improve or even restore the sensitivity of chemotherapy-resistant PC cells to medicines like docetaxel. Clinical trials have been started to assess siRNA’s effectiveness in a variety of solid tumors, either by itself or in conjunction with other treatment drugs (NCT03087591, NCT00672542) [135,136].

## 11. Resistance to Treatments

Treatment resistance remains a significant challenge in PC management. This section discusses various mechanisms of resistance to different therapies.

### 11.1. Androgen Receptor

About 50% of tumors resistant to castration show overexpression of the AR, which can be amplified to boost AR mRNA levels and enhance sensitivity to testosterone. AR mutations are found in 5% to 30% of resistance cases, potentially leading to increased sensitivity. Also, the transformation of antagonists into agonists or better cofactor recruitment (PTEN, PI3K/AKT, FOXA1, AR-V7, GATA2, HOXB13) can lead to resistance [155,156,157].

Moreover, signaling pathways play a role in the development of resistance; for example, PI3K/AKT, RAS-MAPK regulates cell proliferation, c-RAS leads to oncogenesis and progression and elevated levels of MAPK, and c-Src/SFK is highly expressed in CRPC, leading to bone metastasis. The JAK/STAT-3 pathway is in charge of oncogenesis and uses PD-L1 expression to mediate immune function. Also, in 50% of cases, elevated IL-6 levels have been reported [158,159,160].

### 11.2. Chemotherapy

Chemotherapy resistance may result from the activation of the PI3K, protein kinase B (PKB), and mTOR pathways by regulating factors such c-myc and VEGF [161,162].

Bcl2 causes resistance by preventing cytochrome c from being released from mitochondria, which stops the caspase cascade and prevents apoptosis [163]. NF-κB leads to the activation of anti-apoptotic elements such as IAPs and tumor necrosis factor (TNF)-α [164,165]. IL-8 stimulates angiogenesis by overexpressing fibroblast growth factor (FGF) and vascular endothelial growth factor (VEGF) via chemokine receptors 1 and 2 (CXCR1 and 2), causing resistance to chemotherapy [166]. Chemotherapy sensitivity can also be restored by drugs that block pathways such the hedgehog, β-catenin, EGFR, endothelin, and MAPK pathways [167].

Vitamin D shows potential in overcoming docetaxel resistance in prostate cancer, but the risk of hypercalcemia at elevated doses poses a significant challenge to its therapeutic application by using Xe4MeCF3 an analogue of the receptor of vitamin D that inhibits tumor growth in a chemoresistant CRPC patient-derived xenograft model. Current research aims to establish optimal dosages and delivery methods to maximize benefits while reducing risks. It is vital to work closely with healthcare professionals to monitor vitamin D levels and customize treatment plans accordingly, ensuring a balance between therapeutic potential and safety [168].

### 11.3. PARPi Resistance

PARPi resistance can result from mutations in the DNA-binding domains of PARP1 and from mechanisms that enhance the parylation of PARP1 [169].

In BRCA1/2, PALB2, and RAD51C/D, a second mutation of the same gene may restore the reading frame and partial HRR function [170,171,172]. Losing 53BP1 and its effectors (RIF1, REV7/MAD2L2, and shielding complex) may result in the restoration of HRR, causing PARPi resistance [173,174]. Also, nuclease inhibitor HELB loss causes PARPi resistance [175]. Fork protection is achieved even in the absence of functional BRCA2 protein when PTIP or EZH2, which recruit the MRN complex and MUS81, respectively, is lost [176].

Research in mice demonstrates that resistance to olaparib is acquired via activating the P-glycoprotein drug efflux transporter, which lowers PARPis’ cellular availability [177].

## 12. Conclusions

In conclusion, the treatment of PC is complex, and highly individualized, considering each patient’s health, preferences, and cancer stage. Decisions can include active surveillance, surgery, radiation therapy, hormone therapy, and chemotherapy, among other options. The need for an individualized treatment strategy is emphasized by ongoing research advances aimed at maximizing benefits and reducing negative effects. Collaborative interactions between patients and healthcare providers are crucial to achieve the best possible quality of life.

Immunotherapy has emerged as a promising treatment option, particularly for advanced and metastatic cases. The tumor microenvironment plays a critical role in treatment outcomes, with interactions between cancer cells and the immune system contributing to immune evasion and immune suppression and significantly impacting the efficacy of immunotherapies.

For men with metastatic HSPC, especially those with BRCA mutations, relying solely on ADT may not be sufficient. The incorporation of PARPis or chemotherapy can provide additional benefits. In patients without these mutations, chemotherapy options such as docetaxel can also be effective. This customized strategy aims to maximize treatment effectiveness and improve overall survival.

Despite these advances, CRPC remains a challenging condition, highlighting the need for novel therapeutic approaches. The introduction of new treatments including PARPis and other molecular therapies, combined with immunotherapies, offers promising new therapeutic options, as indicated by recent and ongoing clinical trials.

However, several gaps in current research need to be addressed. There is a need for more personalized treatment strategies that consider individual patient characteristics and genetic factors. The optimal sequencing and combination of different treatment modalities, such as immunotherapy with hormone therapy or chemotherapy, require further investigation. Comprehensive genetic and molecular profiling of prostate cancers to identify biomarkers for disease progression and treatment response is also crucial.

Future studies should focus on addressing these gaps, including improvements in imaging techniques for more accurate staging and monitoring, and multidisciplinary research involving oncologists, urologists, radiologists, and immunologists to develop integrated treatment strategies. In this regard, for example, transrectal biopsy was also shown to be safe [178]. Continuous updates from clinical trials and research are essential to incorporate the latest discoveries into clinical practice. While significant progress has been made, the journey to finding a cure for advanced disease is ongoing and requires continued innovation and collaboration.

## Figures and Tables

**Figure 1 cancers-17-00194-f001:**
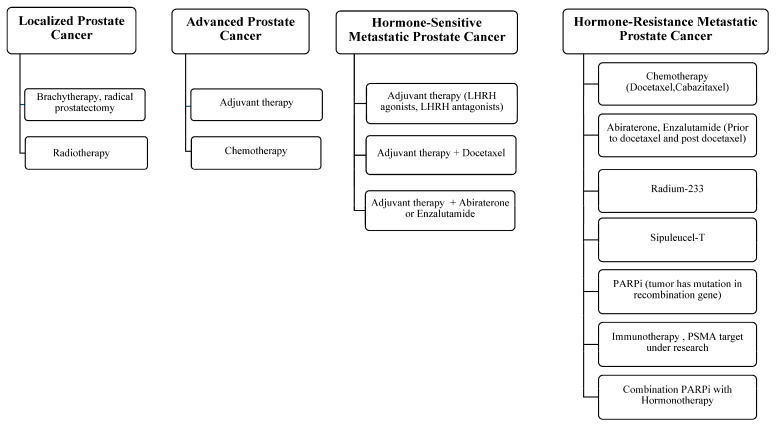
Treatments for prostate cancer.

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
