# Peer review of "Prostate Cancer: A Journey Through Its History and Recent Developments"

_cancers, 2025, doi:10.3390/cancers17020194_

Round 1

Reviewer 1 Report

Comments and Suggestions for Authors

The manuscript "Prostate Cancer: Some Years of History… What is New Now?" by Mallah et al. provides a thorough and engaging review, effectively addressing both the disease biology and therapeutic aspects while also highlighting recent advancements in prostate cancer research.

While the topic is extensively explored in the existing literature, the review would benefit from distinguishing itself by incorporating unique perspectives not typically found in other sources. However, there are some missing pieces of information that should be included, and a few structural modifications are needed in certain sections and subsections, described below: 

1.     Title – The manuscript title should be revised to avoid the use of multiple dots, as it appears awkward. Possible alternatives could be: "Prostate Cancer: A Historical Overview and Current Advances" or "Prostate Cancer: A Journey Through Its History and Recent Developments." The authors may choose a title that best fits the content, but it is recommended to avoid the use of multiple dots.

2.     Simple Summary: The sentence (line no. 8-10; 13-15) is clumsy and unclear, respectively- rephrase it.

3.     In prostate cancer, neuroendocrine prostate cancer (NEPC), whether de novo or castration-induced, is a critical aspect that warrants attention. The author should include a paragraph discussing NEPC, its associated therapeutics, and the strategies being explored to address these challenges.

4.     The author should also incorporate a discussion of the Hedgehog (Hh) signaling pathway in Section 2, "The major signaling pathways" in Prostate Cancer. Hh signaling plays a significant role in disease progression, the advancement of CRPC, and metastasis, both through canonical and non-canonical mechanisms.

5.     Section 3 requires revision for better clarity and structure. The title of Section 3, currently "Different forms of PC," should be changed to more accurately reflect the content. Within this section, only subsection 3.1, "Classification of PC," should be included. Subsections 3.2, "Progression to mCRPC," and 3.4, "Tumor heterogeneity," should be expanded into separate, distinct sections to address these topics in greater detail.

6.     Section 5, titled "Current treatment for PC (Figure 1)," should be revised to remove the mention of the figure in the section title. The reference to the figure should instead be incorporated into the written portion of the section.

7.     The manuscript should follow a consistent writing pattern throughout. If an introduction is provided for one section, it should be included in all sections to maintain uniformity. Additionally, if subsections are not bolded, this formatting should be applied consistently across the manuscript. For example, in Section 5.5.7, "Immunotherapy" is bolded while other subsections are not; this inconsistency should be corrected to align with the journal’s formatting instructions.

8.     It would be helpful to include a summary table in Section 6, titled "Novel agents and clinical trials in PC (2023-2024)," -an overview of both active and inactive clinical trials. The table should include details such as the trial name, phase, status (active/inactive), the type of agents being tested, and any notable results or outcomes.

9.     The conclusion should include a summary of key points regarding immunotherapy in prostate cancer, emphasizing the significance of the tumor microenvironment in treatment outcomes. Additionally, the conclusion should outline the existing gaps in current research and suggest directions for future studies.

Author Response

Please see joined document, responses to comments of Reviewer 1 are in red characters.

Reviewer 2 Report

Comments and Suggestions for Authors

I see no specific improvements to this work to be done.

The manuscript is a thorough analysis and update of the prostate cancer management, a malignancy that is in the top of threatening life malignancies of nowadays.

The scientific approach is relevant, because, offers a useful tool for the practitioners who deal with this challenging malignancy for the management of the disease, especially the therapeutic one.

The added value of this work is the detailed update of the prostate cancer management, no effort in this regard being useless and and having to be made known.

The conclusions are consistent with the evidence and arguments presented.

Finally, I have no other observations for the authors.

Author Response

Please see joined document, responses to comments of Reviewer 2 are in red characters.
